# Highly active and thermostable submonolayer La(NiCo)O$_\Delta$ catalyst stabilized by a perovskite LaCrO$_3$ support

Tingting Zhao [1], Jiankang Zhao[1], Xuyingnan Tao[1], Haoran Yu [1], Ming Li[1], Jie Zeng [1] & Haiqian Wang [1✉]

It is important to develop highly active and stable catalysts for high temperature reactions, such as dry reforming of methane. Here we show a La(NiCo)O$_\Delta$ (LNCO) submonolayer catalyst (SMLC) stabilized by the surface lattice of a perovskite LaCrO$_3$ support and demonstrate a Ni-Co synergistic effect. The submonolayer/support type catalyst was prepared by in-situ hydrogen reduction of a LaNi$_{0.05}$Co$_{0.05}$Cr$_{0.9}$O$_3$ precursor synthesized by a sol-gel method. The LNCO-SMLC is highly active and very stable during a 100 h on stream test at 750 °C under the reaction conditions of dry reforming of methane. The catalyst also shows good anti-coking ability. We found that the synergistic effect between Ni and Co atoms in LNCO-SMLC remarkably improved the thermostability of the catalyst. This work provides a useful concept for designing atomically dispersed catalysts with high thermostability.

[1] Hefei National Laboratory for Physical Science at the Microscale, University of Science and Technology of China, 230026 Hefei, Anhui, People's Republic of China. ✉email: hqwang@ustc.edu.cn

Dry reforming of methane (DRM) is an important reaction that converts $CH_4$ and $CO_2$, two atmospheric greenhouse gases, into valuable syngas ($H_2 + CO$)[1]. DRM is featured by high temperature (700–1000 °C) and reducing reaction atmosphere. Thermodynamically, a few side reactions accompany the main DRM reaction, such as reverse water-gas shift reaction (RWGS, $CO_2 + H_2 = CO + H_2O$), $CH_4$ decomposition reaction ($CH_4 = 2H_2 + C$), and Boudouard reaction ($2CO = CO_2 + C$). Thus, besides providing high activity and selectivity, a DRM catalyst should also be sintering resistant and coking resistant. The harsh reaction conditions of DRM make it a representative probe reaction to evaluate the effectiveness and robustness of a catalyst.

Perovskite oxides with the formula of $ABO_3$ are, in general, thermally stable at elevated temperatures. They are widely used as supports or precursors to fabricate metal-support catalysts[2–4]. Recently, experimental and theoretical studies show that noble-metal single atoms, such as Pt and Au, can be effectively anchored on perovskite supports such as $LaFeO_3$ and $SrBO_3$ (B = transition metals)[5–7]. Moreover, a perovskite itself may be catalytically active[8–10]. Transition metal atoms at the B-site are generally believed to be active owing to the d orbitals, while the A-site metals are less decisive[11]. Suntvich et al. reported that perovskite $Ba_{0.5}Sr_{0.5}Co_{0.8}Fe_{0.2}O_{3-\delta}$ (BSCF) catalyzes the OER with intrinsic activity that is at least an order of magnitude higher than that of the state-of-the-art iridium oxide catalyst in alkaline[12]. They proposed that the $e_g$ orbital of surface transition metal ions participates in σ-bonding with a surface-anion adsorbate, and the $e_g$ filling can greatly influence the binding of reaction intermediates to the oxide surface and thus the activity. Kim et al.[13] demonstrated that $La_{0.9}Sr_{0.1}CoO_3$ has higher activity than a commercial Pt-based catalyst for NO oxidation. It is also reported that perovskite $La_{0.75}Sr_{0.25}Cr_{0.5}Mn_{0.5}O_{3-\delta}$ (LSCM) can be used as an anode material for solid oxide fuel cells and shows high activity for $CH_4$ oxidation[14].

In an ideal perovskite, the larger A-site cations occupy the cubic center, while the smaller B-site cations, octahedrally coordinated to six oxygen anions each, occupy the corner sites with the $BO_6$ octahedra arranged in a corner-shared configuration. Thus, a perovskite can be considered a structure constructed by $BO_6$ superatoms with A-site atoms filled in between. In this sense, a perovskite is a native atomically dispersed catalyst with the catalytically functional B-site atoms embedded in $BO_6$ octahedra and separated by A-site cations. The chemical (e.g., activity and selectivity) and physical (e.g., stability) properties of perovskites can be tuned conveniently by substitutional doping or creating vacancies at both the cation and anion sites.

It is proved that monovalent or low-valent $Ni^{\delta+}$ and $Co^{\delta+}$ are catalytically active owing to the partially occupied 3d orbitals[15–18]. To create catalytically active open sites with exposed Ni 3d orbitals accessible for reactants, we may simply intend to tune the $NiO_6$ superatoms to $NiO_x$ (x < 6) by introducing oxygen vacancies into the Ni-containing perovskite, e.g., $LaNiO_3$. However, reducible perovskites containing catalytic active metal atoms, such as $LaNiO_3$ or $LaCoO_3$, will be over-reduced to metallic Ni or Co nanoparticles supported on $La_2O_3$ (or its derivates) under the harsh DRM reaction conditions. On the other hand, irreducible perovskites, such as $LaCrO_3$, are catalytically inert. Is it possible to stabilize the catalytically active B-site Ni or Co atoms in their low-valent cationic states by preventing the perovskites from over-reducing? The fact that a perovskite can accommodate a large number of oxygen vacancies ($V_O$), especially on the surface, provides the primary possibility. Thus, stabilizing the catalytically active reducible perovskite ($LaNiO_\Delta$ and/or $LaCoO_\Delta$) by stable irreducible perovskite support (such as $LaCrO_3$) is interesting and attractive. In recent work, we

demonstrated that bimetallic Ni–Co catalysts supported on $La_2O_3$–$LaFeO_3$ exhibit higher activity and coking resistance than its monometallic (Ni or Co) counterparts for DRM reactions, which should be essentially attributed to the synergistic effect between Ni and Co[2]. We will show that such a synergistic effect also plays an important role in stabilizing SMLCs.

In this work, we report the construction of a novel submonolayer catalyst: La(NiCo)$O_\Delta$ submonolayer (LNCO-SML) stabilized on the surface of a perovskite $LaCrO_3$ support. We prove that LNCO-SML is highly active and stable at elevated temperatures. The formation and stabilization mechanisms of the LNCO-SML and Ni–Co synergistic effect were discussed by comparative studies between $LaNi_{0.05}Co_{0.05}Cr_{0.9}O_3$ and its single metallic counterparts, $LaNi_{0.1}Cr_{0.9}O_3$ and $LaCo_{0.1}Cr_{0.9}O_3$.

## Results

The $LaNi_{0.05}Co_{0.05}Cr_{0.9}O_3$, $LaNi_{0.1}Cr_{0.9}O_3$, and $LaCo_{0.1}Cr_{0.9}O_3$ catalysts were activated by $H_2$ reduction before being used for DRM reaction. As we will prove in the following content, our samples can be seen as catalytically active SMLs supported on perovskite $LaCrO_3$ grains. For the convenience of discussions, we use the corresponding SMLs to designate the samples. The $LaNi_{0.05}Co_{0.05}Cr_{0.9}O_3$, $LaNi_{0.1}Cr_{0.9}O_3$, and $LaCo_{0.1}Cr_{0.9}O_3$ samples are denoted as LNCO, LNO, and LCO, respectively. We use the suffixes, -F, -R, and -U, to denote the fresh, $H_2$-reduced, and used samples, respectively. The number following the abbreviations of used samples describes the time on stream (TOS) in hours. Thus LNCO-U100 and LCO-U100 represent the used LNCO and LCO samples after 100 h on stream tests, and LNO-U5 and LNO-U24 represent the used LNO samples after 5 h and 24 h on stream tests, respectively.

**Catalytic performance**. The temperature-dependent $CH_4$ conversions over LNCO between 600 and 850 °C (Fig. 1a) reveal that the LNCO catalyst has high activities, which are close to the thermodynamic equilibrium values[19–22] (without carbon deposition) under the specific experimental conditions. The long-term performance of LNCO was performed at 750 °C as shown in Fig. 1b. The $CH_4$, $CO_2$ conversions, and $H_2$/CO ratio over LNCO are 85%, 88%, and 0.90, respectively. These results indicate that the LNCO catalyst effectively promoted the main DRM reaction. The $H_2$/CO ratio is smaller than 1 is due to the RWGS reaction. The effective catalysis of LNCO on the main DRM reaction is also reflected by the high selectivity of $H_2$ and CO (Supplementary Fig. 1), which are 0.94 and 0.99, respectively. Especially, the selectivity of CO is very close to 1 suggesting that side reactions of carbon deposition ($CH_4$-decomposition reaction and Boudouard reaction) hardly happened. The LNCO catalyst is also very stable, and no degradation can be detected during a 100 h on stream test.

For comparison purposes, the conversions and $H_2$/CO ratio over LNO and LCO are also illustrated in Fig. 1b. The conversions of $CH_4$ and $CO_2$ over LNO catalyst are stable at around 85 and 86% in the initial 15 h but quickly decrease to below 10% at 24 h. On the other hand, the conversions of $CH_4$ and $CO_2$ over LCO decrease slowly and continuously from 67 and 78% to 16 and 32%, respectively, during a 100 h on stream test. The conversions of $CH_4$ (Fig. 1a) and $CO_2$ (Supplementary Fig. 1a) over LNCO are in thermoequilibrium. Meanwhile, the $H_2$/CO ratio (0.9) of LNCO is also very close to the thermoequilibrium value (0.92)[21]. It is seen from Fig. 1b that the decrease in the $H_2$/CO ratio of LNO and LCO happened simultaneously with the decrease in the $CH_4$ and $CO_2$ conversions. The decrease of the $CH_4$ and $CO_2$ conversions over LNO and LCO to below the thermoequilibrium value indicates the DRM reaction switches from the thermodynamic-control process

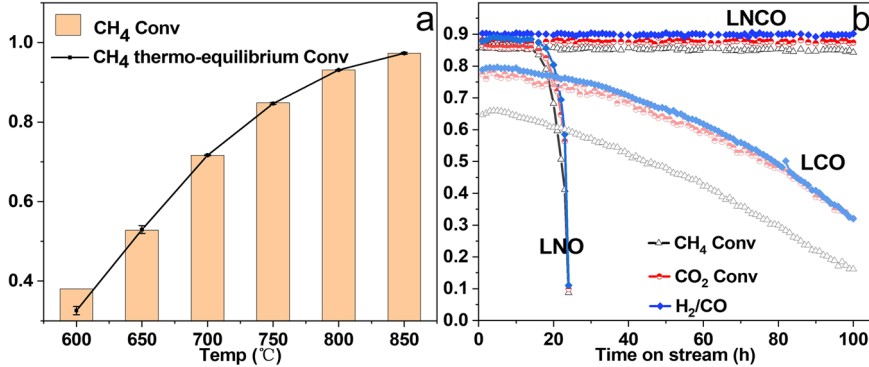

**Fig. 1 DRM performance over LNCO, LNO, and LCO catalysts. a** Temperature-dependent $CH_4$ conversion between 600 and 850 °C over LNCO catalyst, **b** DRM performance over LNCO, LNO, and LCO catalysts at 750 °C. Conditions: $CH_4$:$CO_2$ = 1:1, total flow rate = 60 sccm (GHSV = $1.2 \times 10^4$ mL $g_{cat}^{-1}$ $h^{-1}$). LNCO, LNO, and LCO represent $LaNi_{0.05}Co_{0.05}Cr_{0.9}O_3$, $LaNi_{0.1}Cr_{0.9}O_3$, and $LaCo_{0.1}Cr_{0.9}O_3$ catalysts, respectively.

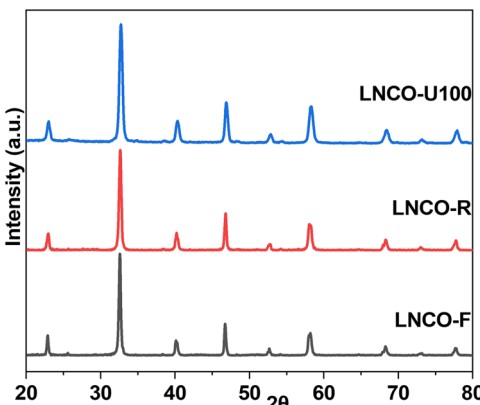

**Fig. 2 XRD patterns of perovskite LNCO samples.** The XRD patterns are normalized. LNCO-F, LNCO-R, and LNCO-U100 represent fresh, $H_2$-reduced, and used after 100 h time on stream $LaNi_{0.05}Co_{0.05}Cr_{0.9}O_3$ samples, respectively.

into a kinetic-controlled process. It is reported that Ni is highly active for RWGS reaction, and the RWGS is very fast. It immediately reaches thermoequilibrium in a wide temperature range (e.g., above 500 °C, depending on the contact time, etc.)[23,24]. Or in other words, the RWGS is less sensitive to the catalysts' activity at high temperatures. In our experimental conditions (750 °C), the rapid decrease in the $H_2$/CO ratio of LNO and LCO should be dominated by the degradation of the catalyst, which reduces the amount of DRM-generated $H_2$ and CO. In this case, the less catalyst-dependent RWGS reaction will consume a relatively larger proportion of the total $H_2$ and add a more significant amount of CO to the total. Thus, LNO and LCO show a rapid decrease in the $H_2$/CO ratio during the reaction. On the other hand, the LNCO could suppress the RWGS reaction very well is because the catalyst is stable and the conversions of $CH_4$ and $CO_2$ and $H_2$/CO ratio maintain at a high level.

The different performance of LNCO, LNO, and LCO indicates that the synergistic effect between Ni and Co in LNCO not only keeps the high activity but also makes the catalyst thermostable at high temperatures. We also synthesized NiCo@LaCrO₃ with the same nominal composition to LNCO by preparing the LaCrO₃ support first and then depositing Ni/Co ions by incipient wetness impregnation. However, the catalytic performance is very different from LNCO, LNO, and LCO prepared by the sol-gel combustion method. The conversions of $CH_4$ and $CO_2$ over NiCo@LaCrO₃ are lower than LNCO and decrease fast in a

slowdown trend (See Supplementary Fig. 2), which is in accord with the sintering degradation behavior of metal-support catalysts[25]. As a comparison, the conversions of $CH_4$ and $CO_2$ over LNO and LCO catalysts decrease in an accelerated way with time on stream. Such a NiCo@LaCrO₃ metal-support catalyst is less stable than LNCO.

The high catalytic activity and very stable long-term performance differ the LNCO catalyst from usual metal-support catalysts reported in literatures[26–30], especially if we consider the relatively low Ni loading (1.3 wt%) and low specific surface area (10.0 $m^2$ $g^{-1}$) of LNCO-F. As a relevant example, we reported in a recent work that the $CH_4$ and $CO_2$ conversions over the metal-support type catalyst, Ni–Co/La₂O₃-LaFeO₃ with 7.2–10.8 wt% Ni loading and ~10 $m^2$ $g^{-1}$ specific surface area, are 70 and 80%, respectively, at 750 °C under the same DRM conditions[2], which is much lower than those over LNCO. Moreover, the degradation behavior of LNO and LCO cannot be explained by metal-support catalysts. The conversions of $CH_4$ and $CO_2$ over LNO and LCO catalysts decrease in an accelerated way with time on stream. In contrast to the LNO and LCO, the decrease in conversions of $CH_4$ and $CO_2$ over NiCo@LaCrO₃ shows a slowdown trend (See Supplementary Fig. 2), which is in accord with the sintering degradation behavior of metal-support catalysts[25]. The high catalytic activity of LNCO implies that the number of active sites or the dispersion of Ni and Co on the surface of the catalyst is high. The long-term stable performance at 750 °C suggests that the LNCO catalyst is thermostable, thus catalyst sintering and carbon deposition should be small.

**Crystalline structure and reducibility.** The XRD pattern of LNCO-F is dominated by a well-defined perovskite phase with space group Pbnm (LaCrO₃: JCPDS 71-1231) (Fig. 2). No impurities, such as NiO, Co₂O₃, Co₃O₄, etc. can be detected, indicating that nickel and cobalt atoms are well incorporated into the crystalline perovskite lattice. After LNCO-F undergoes $H_2$ reduction and DRM test, the diffraction patterns and peak positions of LNCO-R and LNCO-U100 samples are almost the same as those of LNCO-F. No separated Ni, Co, or NiCo alloy phases can be detected in LNCO-R and LNCO-U100 should be because the amount of the metallic phases is very small and below the detection limit of XRD. The XRD patterns of LNO and LCO samples (Supplementary Fig. 3) are similar to those of LNCO, in which all the patterns are dominated by the perovskite phase with space group Pbnm (LaCrO₃: JCPDS 71-1231). The XRD single-phase nature of the used sample indicates that the perovskite crystalline structure of LNCO, LNO, and LCO is very stable, and it is difficult to exsolve the Ni and Co atoms from the perovskite matrix under DRM conditions.

Supplementary Fig. 4 shows the TG and DTG profiles of LNCO-F, LNO-F, and LCO-F measured by TPR. The overall mass losses of all the samples are less than 1.0% as the temperature increases from 30 to 1200 °C, indicating that the samples are very stable under hydrogen reduction. Nevertheless, a reduction step at about 300–400 °C is resolved in each of the mass-loss profiles, which can be assigned to the reduction of $Ni^{3+}$ to $Ni^{2+}$ and/or $Co^{3+}$ to $Co^{2+}$ [2,31–34]. The mass losses at 300–400 °C are about 0.06% for LNCO-F, 0.26% for LNO-F, and 0.06% for LCO-F, which are smaller than the theoretical value of 0.33% for a full reduction to $Ni^{2+}$ and $Co^{2+}$, indicating that not all the $Ni^{3+}$ and $Co^{3+}$ cations are reduced to $Ni^{2+}$ and $Co^{2+}$ in this reduction step. The reduction step at about 500 °C should be assigned to the reduction of $Cr^{6+}$ to $Cr^{3+}$ [35] because of the existence of a small amount of $Cr^{6+}$ cations in the fresh samples, as evidenced by XPS (Supplementary Fig. 6). It is interesting to note that no relevant reduction step of $Ni^{2+}$ to $Ni^0$ and/or $Co^{2+}$ to $Co^0$ can be observed, suggesting that the $Ni^{2+}$ and $Co^{2+}$ ions in LNCO, LNO, and LCO are stable and hard to be reduced to $Ni^0$ and $Co^0$. As the temperature increases to above 500 °C, LNO-F further reduces slowly and continuously. This should be related to the slow release of $Ni^{3+}$ from the inside of the perovskite grains and, probably, further reduction of $Ni^{2+}$ to its lower valence state $Ni^{\delta+}$ or $Ni^0$, especially near the surface regions. Similar TPR profiles were also reported by Stojanović et al., and the authors concluded that $LaNi_xCr_{1-x}O_3$ compounds with x < 0.5 do not reduce to nickel metal in the $H_2$ atmosphere at <900 °C [36].

**Surface Ni distribution and electronic interactions**. The surface distribution of Ni and Co is essential since it reflects the number of active sites. We use the Cr 3s XPS spectrum to determine the surface content of Ni and Co because the most intensive Ni $2p_{3/2}$ peak overlaps heavily with La $3d_{3/2}$. Figure 3 shows that the Cr 3s spectrum of LNCO has two well-separated peaks at about 74.4 eV (denoted as $Cr_{S+1/2}$) and 78.6 eV (denoted as $Cr_{S-1/2}$) due to intra-atomic multiplet splitting [37]. They are close to the Ni 3p peak (BE ≈69.5 eV) and Co 3p peak (BE ≈60.5 eV) and thus can be used as an internal reference to determine the Ni and Co contents. Both Ni and Co contents show a decreasing trend with the sample sequence of LNCO-F → LNCO-R ≈ LNCO-U100 (see Fig. 3 and Supplementary Table 1). The Ni and Co contents detected by XPS depend on the probing depth (2–5 nm for solids [38]) and the distribution form of Ni and Co. In LNCO-F, Ni and Co atoms are distributed homogeneously within the

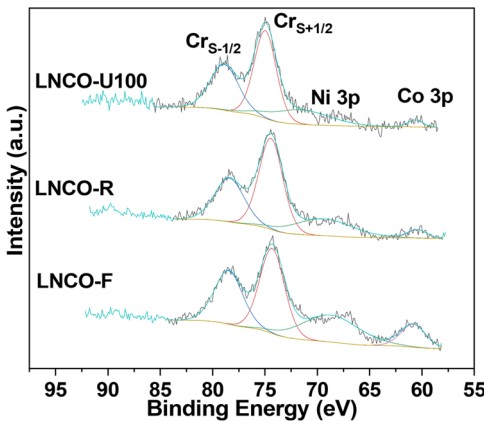

**Fig. 3 Ni 3p, Co 3p, and Cr 3s XPS spectra of LNCO samples.** $Cr_{S+1/2}$ and $Cr_{S-1/2}$ denote intra-atomic multiplet splitting of Cr 3s with the remaining 3s electron coupled parallel and antiparallel to the 3d electrons, respectively. LNCO-F, LNCO-R, and LNCO-U100 represent fresh, $H_2$-reduced, and used after 100 h time on stream $LaNi_{0.05}Co_{0.05}Cr_{0.9}O_3$ samples, respectively.

perovskite lattice (see Supplementary Fig. 7), and thus XPS will "see" all the Ni and Co atoms within the probing depth. Once the Ni and Co atoms in the surface layer aggregate into 3D nanoparticles with diameters larger than the probing depth, the Ni and Co contents detected by XPS will reduce. In the case of the coverage of Ni and Co nanoparticles are small in terms of projected area on the perovskite surface (fewer and larger NiCo 3D nanoparticles, see Supplementary Figs. 8 and 9), the surface Ni and Co contents detected by XPS mainly depend on the atomically dispersed Ni and Co atoms on the perovskite surface. Thus, the XPS results indicate that the number of atomically dispersed Ni and Co atoms on the surface of LNCO-R is smaller than that of LNCO-F but close to or slightly larger than that of LNCO-U100. The decrease in the Ni and Co contents also implies a transverse migration of the atomically dispersed Ni and Co atoms when they aggregate into 3D nanoparticles.

The Cr 3s XPS spectra also reflect the interaction between the B-site Ni, Co, and Cr atoms. The multiplet splitting of Cr 3s arises from the exchange interaction between the spin of the remaining Cr 3s electron and local unpaired valence 3d electrons, and the magnitude of the energy separation $\Delta E = (2S + 1) \, G^2(3s, 3d)$, where S is the total spin of the local valence 3d electrons, and $G^2(3s, 3d)$ the exchange integral between the 3s and 3d orbitals [38–42]. For the specific valence state of $Cr^{3+}$, $\Delta E$ also depends on the electronegativity of the ligand because the covalency between $Cr^{3+}$ and ligand ($O^{2-}$ in the present case) affects the overlap (or the exchange integral) between the 3s and 3d orbitals [41].

Supplementary Table 1 shows that $\Delta E$ reduces from 4.2 eV for LNCO-F to 4.0 and 3.9 eV for LNCO-R and LNCO-U100, respectively. The similar changing trend of $\Delta E$ and the Ni and Co contents (see Fig. 3 and Supplementary Table 1) with the sample sequence (LNCO-F, LNCO-R, and LNCO-U100) suggests that there are some kinds of interactions between the B-site Cr and Ni or Co atoms. For Cr compounds ($LaNi_{0.05}Co_{0.05}Cr_{0.9}O_3$ in the present case), as a first-order approximation, $\Delta E$ depends on the total 3d ground-state spin [38,40,43]. The observed $\Delta E$ (around 4 eV) for the LNCO samples agrees well with the reported values for Cr(III) compounds [40,41,43], confirming that the Cr ions are in the $Cr^{3+}$ oxidation state. The very small amount of $Cr^{6+}$ in LNCO-F (see Supplementary Fig. 6) does not contribute to the exchange splitting because S = 0 for the $3d^0$ electron configuration. The larger $\Delta E$ observed in LNCO-F should be induced by the ligand effect [41]. Charge transfer from $Ni^{3+}$ (3d) or $Co^{3+}$ (3d) through $O^{2-}$ (2p) to $Cr^{3+}$ (3d) will push the Cr 3d orbital inward (more localized), thus increasing the overlap (the exchange integral) between the Cr 3s and 3d orbitals. In this sense, Ni or Co can be considered as an electron donor of $LaCrO_3$. According to Saitoh's work [44], the doping electrons in $LaCrO_3$ have more Cr 3d character and less O 2p character, which means that the doped electrons from Ni or Co mainly go to the Cr 3d orbitals. Thus, the decreases of $\Delta E$ in the LNCO-R and LNCO-U100 samples can be explained by the decrease in the atomically dispersed Ni and Co contents because fewer $Ni^{3+}$ and $Co^{3+}$ ions are available for the charge transfer. Nevertheless, the very close $\Delta E$'s between the LNCO-R and LNCO-U100 samples indicate that the atomically dispersed Ni and Co atoms are very stable over the 100 h on stream test.

Similar XPS features were also observed in LNO and LCO (see Supplementary Fig. 5 and Supplementary Table 1). Therefore, the above discussions can also be applied to LNO and LCO.

**Structure and composition of SMLs**. Typical HR-TEM images of the LNCO samples (Fig. 4) reveal that the catalyst is composed of well-crystallized grains, and the spacing between the fringes of the

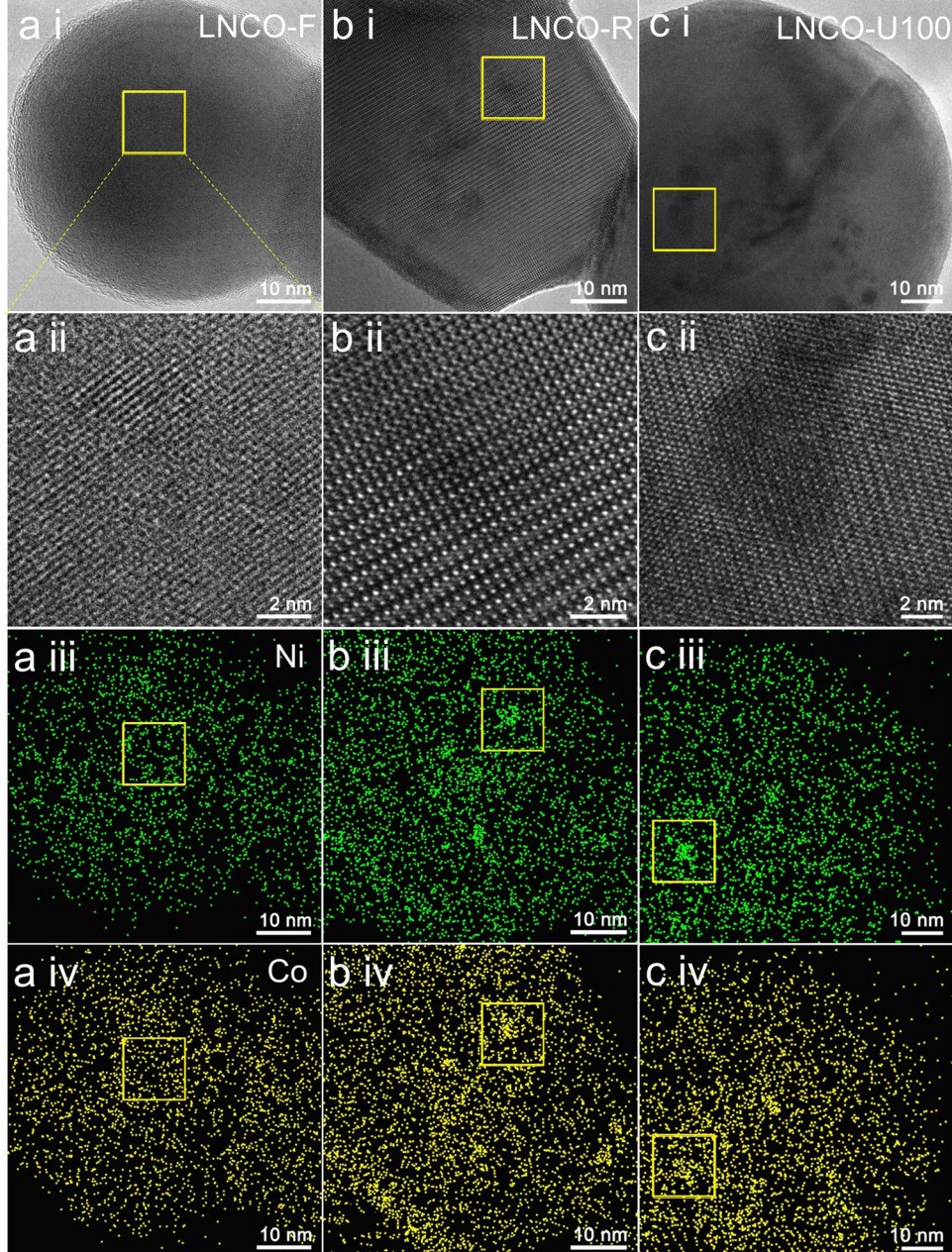

**Fig. 4 HR-TEM images and EDS Mapping of Ni and Co of LNCO samples. a** LNCO-F, **b** LNCO-R, and **c** LNCO-U100 represent fresh, $H_2$-reduced, and used after 100 h time on stream $LaNi_{0.05}Co_{0.05}Cr_{0.9}O_3$ samples, respectively. The 1st row: HR-TEM images, the 2nd row: magnified HR-TEM images corresponding to the yellow squares in the 1st row, the 3rd row: EDS Mapping of Ni corresponding to the HR-TEM images in the 1st row, and the 4th row: EDS Mapping of Co corresponding to the HR-TEM images in the 1st row. The regions marked by yellow squares in (**b**, **c**) are Ni-rich and Co-rich.

grains can be indexed to the perovskite phase as determined by XRD. EDS mapping of LNCO-F shows homogeneous distributions of all the La, Cr, Ni, Co, and O elements, indicating that Ni and Co atoms dissolve uniformly in the $LaCrO_3$ perovskite lattice (Fig. 4 and Supplementary Fig. 7). LNCO-R and LNCO-U100 differ from LNCO-F in that we can observe many dark regions (marked by yellow square) in the HR-TEM images (Fig. 4bi and 4ci) and the dark regions are Ni- and Co-rich (denoted as NiCo-rich regions) (Fig. 4biii, biv, ciii, and civ). It is interesting to note that these NiCo-rich regions share the same lattice structure with the $LaCrO_3$ grains, indicating that the Ni and Co atoms occupy the perovskite lattice sites, but do not exist in the form of metallic Ni–Co alloy nanoparticles. On the other hand, no aggregation of La occurs in LNCO-R and LNCO-U100 (see Supplementary

Figs. 8 and 9), suggesting that La distributes uniformly in these samples but did not segregate into the $La_2O_3$ phase or its derivatives. This further confirms that the NiCo-rich regions are, essentially, perovskite-type $LaNiCoO_\Delta$ (LNCO). No protrusions can be found at the projected boundaries of the grains in the LNCO-R and LNCO-U100 samples (Fig. 4) suggesting that the LNCO is in the two-dimensional (2D) form. This resulted from the reducible nature of LNCO. As we know, 3D bulk perovskite $LaNiO_3$ and $LaCoO_3$ are not stable and will decompose completely into metallic Ni and Co nanoparticles and $La_2O_3$ under $H_2$-reduction and DRM conditions[2]. The stable existence of submonolayer LNCO should be primarily attributed to the strong interaction between the LNCO and the $LaCrO_3$ support. Moreover, the overlap of the Ni-rich and Co-rich regions in Fig. 4

indicates that the interactions between the B-site Ni and Co atoms make LNCO more stable than individual LaNiO$_\Delta$ or LaCoO$_\Delta$. Thus, we know that the synergistic effect between Ni and Co further stabilizes the LNCO. We believe that the LNCO submonolayer is V$_O$-rich based on the understanding that there are plenty of oxygen vacancies on the surface of reducible perovskites. Such LNCO-SML stabilizes the catalytically active Ni and Co atoms in their low-valent states (Ni$^{\delta+}$ and Co$^{\delta+}$). Thus, the LNCO-SMLs should be responsible for the high catalytic activity of the LaNi$_{0.05}$Co$_{0.05}$Cr$_{0.9}$O$_3$ catalyst. We can also observe some metallic NiCo nanoparticles with lateral sizes of more than 50 nm in the LNCO-U100 sample (see Supplementary Fig. 9b). However, the number of NiCo nanoparticles per unit area on the surface is very low as compared with that in the traditional metal-support catalysts[2]. The few but large NiCo nanoparticles cannot explain the very high catalytic activity of the LNCO-U100 sample. Thus, we conclude that the high catalytic activity of LNCO-U100 comes from the LNCO-SMLs but not from the metallic NiCo nanoparticles.

To further confirm that catalysts are of SMLs/support type and the SMLs are responsible for the high activity, we also performed similar HR-TEM and EDS-Mapping analyses on the LNO samples (see Fig. 5 and Supplementary Figs. 10–13). As expected, the same SML features, including the dark regions in HR-TEM, the Ni-rich regions in the EDS-Mapping images, and the sharing of the lattice with the LaCrO$_3$ support, are represented in the LNO-R and LNO-U5 samples. Thus, we can assign the SMLs to LaNiO$_\Delta$ submonolayers (LNO-SMLs). Such an LNO-SMLs stabilizes the catalytically active Ni atoms in their low-valent

states Ni$^{\delta+}$. Thus, the LNO-SMLs should be responsible for the initial catalytic activity of the LNO catalyst. Few LNO-SMLs and more Ni nanoparticles (see Fig. 5 and Supplementary Fig. 13) in the deactivated LNO-U24 sample further prove that the high catalytic activity of LNO-U5 (see Fig. 5 and Supplementary Fig. 12) comes from the LNO-SMLs but not from the metallic Ni nanoparticles. Similar features can also be found in the LCO-U100 sample (Supplementary Fig. 14). In contrast, Supplementary Fig. 15 shows that the used NiCo@LaCrO$_3$ sample is dominated by metallic Ni nanoparticles.

To understand mechanisms for the formation of the SMLs on the LaCrO$_3$ surface, we examined the HR-TEM (Fig. 6a) and Z-contrast HAADF images of LNCO-R (Fig. 6b, c) in the same selected area. It is interesting to see from Fig. 6b that there are some dark regions at the margins of SMLs (bright regions), which should be resulted from missing La, Ni, and Co atoms (NiCo-deficient regions). This phenomenon implies that the SMLs are formed by short-distance transverse diffusion of Ni, Co, and La atoms on the LaCrO$_3$ surface. The line profiles across the SMLs (Fig. 6d) reveal that both A- and B-site elements show higher intensity in the LNCO-SML regions, further confirming that SMLs are La(NiCo)O$_\Delta$. The pronounced difference in the intensities between the SML and the NiCo-deficient regions in the line profile (Fig. 6d) should result from both the geometric and atomic factors: (1) the SMLs are at least one atomic step higher than the NiCo-deficient regions due to the out-migration of Ni/Co and La atoms, and (2) the atomic numbers of Ni and Co are higher than Cr, thus Ni and Co appear a little brighter in the HAADF images. Similar phenomena can also be observed in the

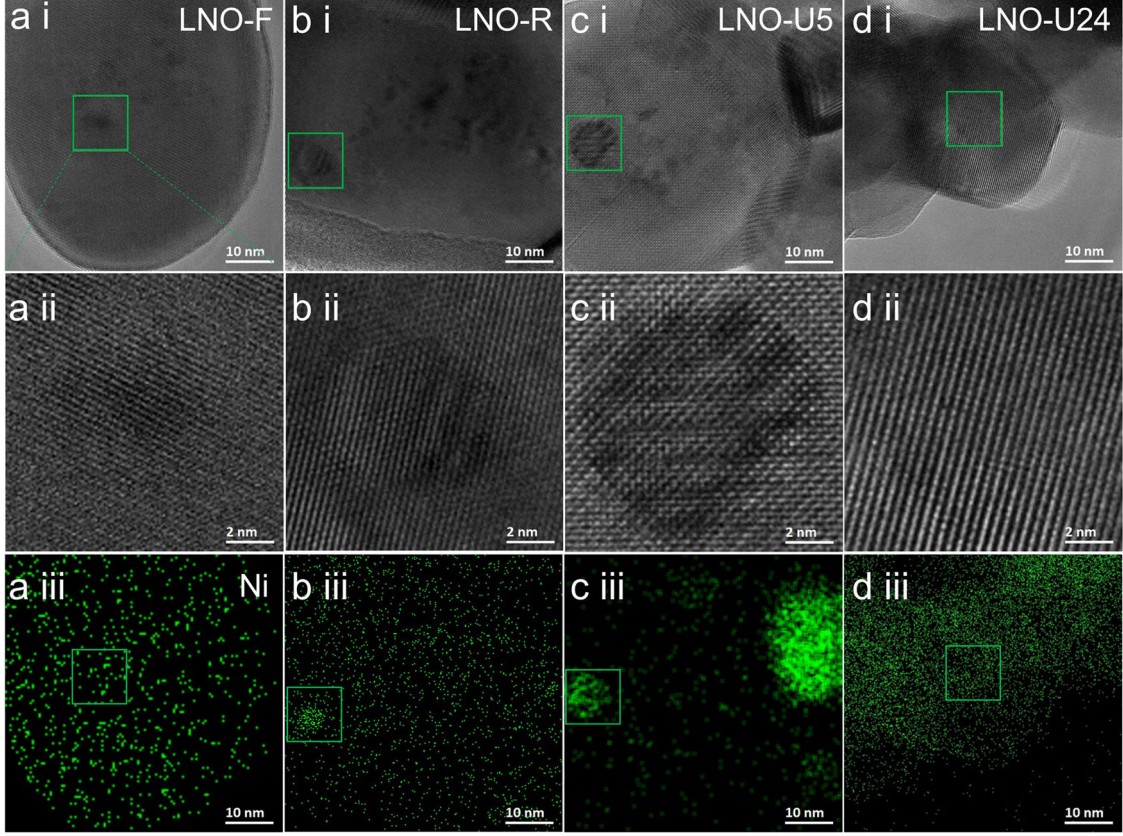

**Fig. 5 HR-TEM images and EDS Mapping of Ni of LNO samples. a** LNO-F, **b** LNO-R, **c** LNO-U5, and **d** LNO-U24 represent fresh, H$_2$-reduced, used after 5 h time on stream and used after 24 h time on stream LaNi$_{0.1}$Cr$_{0.9}$O$_3$ samples, respectively. The 1st row: HR-TEM images, the 2nd row: magnified HR-TEM images corresponding to the green squares in the 1st row, and the 3rd row: EDS Mapping of Ni corresponding to the HR-TEM images in the 1st row. The regions marked by green squares in (**b**, **c**) are Ni-rich.

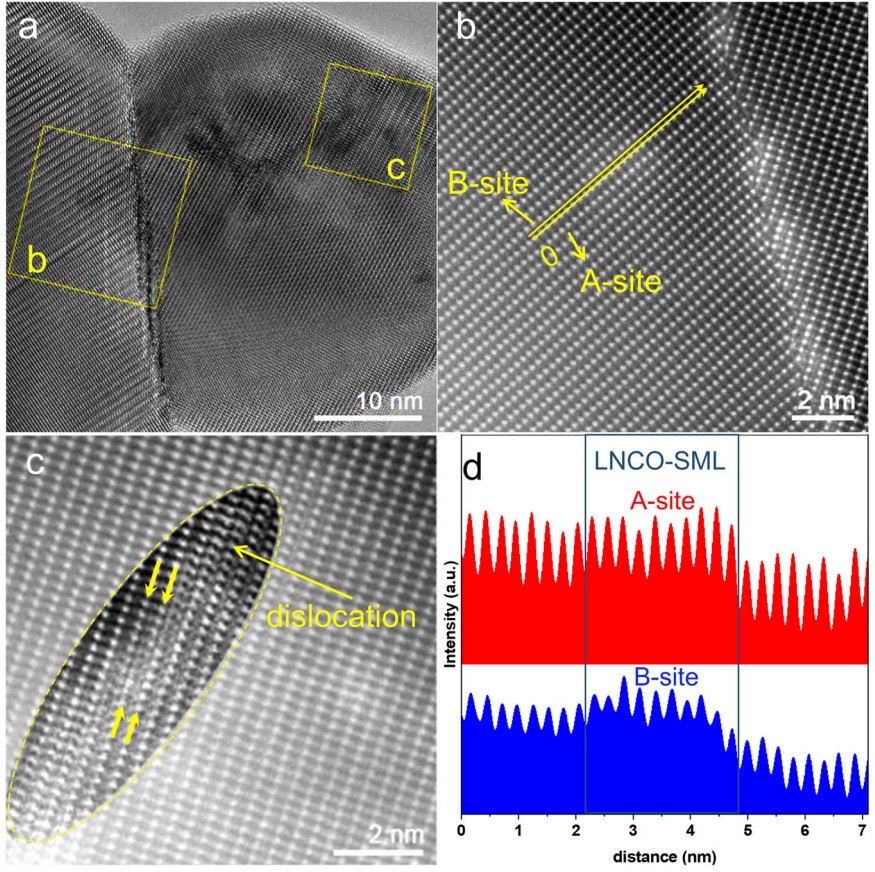

**Fig. 6 HR-TEM and HAADF images of LNCO-R sample. a** HR-TEM, **b**, **c** HAADF images of LNCO-R sample corresponding to the yellow squares in **a**. **d** The line profile across A- and B-site elements as indicated in **b**. The contrast in the ellipse in **c** was enhanced to highlight the dislocation in LNCO-SML. The surface direction is [020]. LNCO-R represents $H_2$-reduced $LaNi_{0.05}Co_{0.05}Cr_{0.9}O_3$ sample. LNCO-SML represents $La(NiCo)O_\Delta$ submonolayer.

LNO-R sample (Supplementary Fig. 16), confirming that the SMLs in LNO-R are $LaNiO_\Delta$ submonolayers.

It is also interesting to note that the SMLs tend to reside at the defect sites, such as grain boundaries (Fig. 6b) and dislocations (Fig. 6c) in the LNCO-R sample. Such kinds of solid defects interrupt the periodicity of the surface lattice, and thus, block the surface migration of Ni and Co single atoms. The presence of these defects not only enhances the anchoring of SMLs on the $LaCrO_3$ surface but could also alter the electronic structures of the active Ni and Co single atoms in the catalyst. The same features of SML reserves in LNCO-U100 (Supplementary Fig. 17) indicate that the SMLs are stable during the 100 h long-term DRM test. In contrast, the metallic NiCo nanoparticles grow much larger after 100 h on stream test due to Ostwald ripening (see Supplementary Fig. 9b). On the other hand, no dislocations nor aggregation of SMLs at the grain boundaries can be observed in the less stable LNO-R sample (see Supplementary Fig. 16), proving that the Ni–Co synergistic effect plays an important role in stabilizing the SMLs.

## Discussion

**The proposed model for SMLC: activity and stability.** Based on the above experimental observations and analyses, we propose a simplified model to qualitatively illustrate the formation and catalytic activity of SMLs on the stable low index [020] surface (viewing from the [100] direction) of a perovskite $LaCrO_3$ as schematically shown in Fig. 7. We start from an ideal picture that a monolayer $LaMO_3$ (M = Ni or Co) is epitaxially grown on the surface of $LaCrO_3$ (see the $MO_6$ octahedron marked by a gray rectangle in Fig. 7). Apparently, O atoms at the six corner sites of

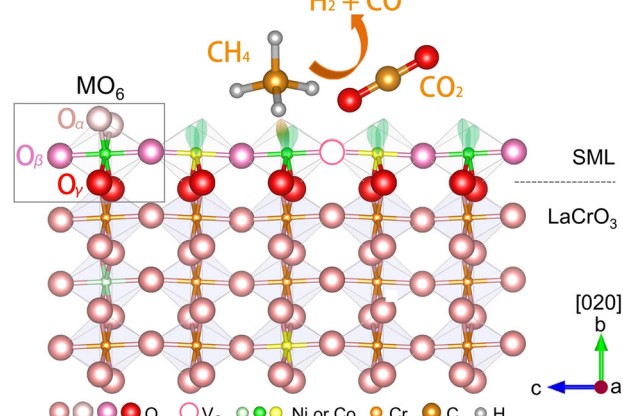

**Fig. 7 Schematic diagram of LNCO-SML stabilized by perovskite $LaCrO_3$ surface lattice (space group Pbnm, La not shown).** LNCO-SML represents $La(NiCo)O_\Delta$ submonolayer. M represents Ni or Co. The $MO_6$ octahedron in the gray rectangle represents a $Ni^{3+}$ or $Co^{3+}$ cation coordinated to six $O^{2-}$ anions. $O_\alpha$, $O_\beta$, and $O_\gamma$ have different bonding configurations with M and Cr. $CH_4$ and $CO_2$ are the reactants, and $H_2$ and CO are the products of DRM. Losses of $O_\alpha$ and $O_\beta$ will reduce $Ni^{3+}$ or $Co^{3+}$ cations to the low-valent states ($Ni^{\delta+}$ or $Co^{\delta+}$) and expose the Ni or Co 3d orbitals to the reactants (or reaction intermediates).

the $MO_6$ octahedron are not equivalent. The two O atoms located at the top of the octahedron (labeled as $O_\alpha$) have the lowest coordination number to M ($O_\alpha$–M) and should be the least stable ones. The two $O_\beta$ atoms at the middle of the octahedron coordinate to two M atoms each (M–$O_\beta$–M), should be more stable

than $O_\alpha$ due to the increased coordination number. $O_\gamma$, located at the bottom of the octahedron, directly connects to the surface of $LaCrO_3$ with the bonding configuration of M–$O_\gamma$–Cr. $O_\gamma$ should be the most stable ones owing to the strong bonding with less electronegative $Cr^{3+}$.

Upon $H_2$ reduction, $O_\alpha$ ions are removed from the lattice, leaving two electrons behind, and simultaneously, $M^{3+}$ ions are reduced to lower valence states ($M^{2+}$ or $M^{\delta+}$). Once this happens, M 3d orbitals are open to the reactants, and the catalyst becomes activated (Fig. 7). In the process of catalyzing a DRM reaction, it is very likely that the metastable $O_\beta$ is active and involved in the elementary reactions. It is widely accepted that many perovskites are characterized by active lattice oxygen that may take part in the elementary reactions of DRM[31,45]. This means that $O_\beta$ should be exchangeable with the oxygen species deduced from $CO_2$. Such kind of active oxygen helps activate $CO_2$ and suppress carbon deposition. Certainly, there are possibilities that $O_\beta$ escapes from the lattice site and introduces additional oxygen vacancies ($V_O$ in Fig. 7), and thus, further reduces $M^{2+}$ to $M^{\delta+}$ ($0 < \delta < 2$). Many reported works correlated the active catalytic sites to $Ni^{\delta+}$ for Ni-based catalysts[15,16,46]. Yang et al.[15] proved that monovalent Ni(I) atoms anchored on the N-doped graphene matrix are catalytically active owing to the partially occupied 3d orbitals. However, Ni(I) ions are frequently thermally unstable[15,46]. In the present work, the stable $O_\gamma$ in SML should be responsible for the stabilization of the Ni(I) (or $Ni^{\delta+}$). In our case, we proposed that the high activity of LNCO comes from the atomically dispersed low-valent $Ni^{\delta+}$ (and $Co^{\delta+}$) metal cations with open 3d orbitals and filled $e_g$ electrons. Considering that Ni is more active than Co for DRM, we think the catalytic activity of LNCO is dominated by Ni, while Co mainly contributes to enhancing the stability of the submonolayer structure. Moreover, the A-site $La^{3+}$ and charger transfer between Ni, Co, and Cr should also facilitate the anchoring of SML on $LaCrO_3$ support[47]. It is reported that Pt atoms were firmly anchored on the surface of SiC nanocrystallites by forming a 2D superstructure[48]. In analogy, Ni and Co atoms in the 2D SML superstructure should be more stable than separated single Ni or Co atoms on the surface. The increased mixing entropy of Ni–Co in the submonolayer should be also helpful for the stabilization of LNCO-SML.

**Synergistic effect between Ni and Co**. The LNCO catalyst shows higher thermostability as compared to LNO and LCO (see Fig. 1b). It is known that $LaCoO_3$ is more stable than $LaNiO_3$ in reducing atmosphere[49] and the affinity of Co to oxygen is stronger than Ni[2]. In the 2D case, the stronger affinity of Co to oxygen should help stabilize $O_\beta$ in the Co–$O_\beta$–Ni bonding configuration (see Fig. 7) and prevent the LNCO-SML from breaking up. Similarly, a stronger Co–$O_\gamma$ bond should facilitate the anchoring of LNCO-SML to the surface lattice of $LaCrO_3$. The coexistence of Co and Ni in LNCO-R also favors the formation of solid defects (point defects and dislocations etc.) owing to the different ionic radii in reducing atmosphere. It is known that solid defects could facilitate the anchoring of single-atom sites by tuning their surrounding electronic structure and coordination environment[50,51]. Thus, the solid defects as we have seen in Fig. 6 (grain boundaries and dislocations) further enhance the stability of the LNCO-SML.

To understand the superior performance of DRM on Ni–Co/$LaCrO_3$ catalyst, we performed density functional theory (DFT + U) calculations for the stability of Ni and Co atoms on our model catalysts for a few typical configurations. The calculation results show that the configuration with B-site neighboring Ni and Co atoms on the top layer (010 surface) is 0.53 eV more stable than the configuration with one Ni atom on the top surface and one Co atom in the bulk (Supplementary Fig. 18). The configuration with B-site neighboring Co, Ni, Co atoms on the top layer

(001 surface) is 0.14 eV more stable than that with two Co atoms on the top surface and one Ni atom in the bulk (Supplementary Fig. 19). These results indicate that the configuration with both Ni and Co on the surface is more stable than those with only Ni or only Co atom on the surface, and the formation of the Ni–O-Co chain on the surface enhances the stability of La(NiCo)$O_\Delta$-SML supported on $LaCrO_3$, namely the LNCO sample.

**Anti-coking property**. As we have seen in Fig. 1, the catalytic performance of LNCO is stable over the long-term 100 h on stream test, and it is hard to find any filamentous carbon in the TEM images (See Supplementary Fig. 9). These results indicate that the LNCO-SMLs have a good anti-coking property. Kim et al.[52] studied the effect of Ni particle size on coking for DRM reaction and found that Ni particles smaller than 7 nm can effectively suppress the formation of filamentous carbon and highly dispersed metal particles showed a remarkably low coking rate. It is reported that the absence of adjacent active metal atoms (Ni or Co) could prevent the C–C coupling or coking. Such a mechanism was used to explain the coking resistance of Fe single atoms embedded in silica matrix for the high-temperature direct conversion of methane to ethylene[53]. It is also reported that Ni single-atom catalysts favor partial $CH_4$ dehydrogenation and thus the complete decomposition to C is suppressed[1]. The higher oxygen affinity of Co also favors enhancing the coking resistance[2].

Nevertheless, we did detect a small amount of deposited carbon in the LNCO-U100, LNO-U24, and LCO-U100 samples. Our TPO analysis shows the average carbon deposition rates of LNCO-U100, LNO-U24, and LCO-U100 are 0.87, 0.25, and 0.09 $mg_C$ $g_{cat}^{-1}$ $h^{-1}$, respectively (Supplementary Fig. 20). Considering the coexistence of SMLs and a few large Ni(Co) nanoparticles (>50 nm), the deposited carbon in the SML/$LaCrO_3$ catalysts should be correlated to the metallic Ni(Co) nanoparticles. Thus, we tend to conclude that the SMLs not only have high catalytic activity and thermostability but also have a good anti-coking property.

**Conclusions**. We successfully constructed a novel SMLC by in situ hydrogen reduction of a perovskite $LaNi_{0.05}Co_{0.05}Cr_{0.9}O_3$ precursor synthesized by the sol-gel method. The LNCO-SML is highly active and very stable over 100 h on stream test at 750 °C under DRM conditions. We demonstrated that the atomically dispersed Ni and Co atoms in LNCO-SMLC stabilized by irreducible perovskite $LaCrO_3$ hold the high activity of LNO with remarkably improved thermostability. The synergistic effect between Ni and Co further enhances the anchoring of LNCO-SMLs on the surface lattice of a perovskite $LaCrO_3$ support by introducing stronger Co–O bonds and more solid defects.

This work provides a useful concept for designing atomically dispersed catalysts with high thermostability. There is plenty of room to explore SMLCs with a large variety of catalytic active noble or non-noble-metal atoms for other reactions.

## Methods

**Catalyst preparation**. $LaNi_{0.05}Co_{0.05}Cr_{0.9}O_3$ (LNCO), $LaNi_{0.1}Cr_{0.9}O_3$ (LN), and $LaCo_{0.1}Cr_{0.9}O_3$ (LC) catalyst precursors were synthesized using a sol-gel self-combustion method[2,54]. All the chemicals of analytical grade were purchased from Sinopharm Chemical Agent Company. Lanthanum oxide ($La_2O_3$) was entirely dissolved in nitric acid aqueous solution. The stoichiometric ratio of nickel nitrate hexahydrate ($Ni(NO_3)_2 \cdot 6H_2O$) and/or cobalt nitrate hexahydrate ($Co(NO_3)_2 \cdot 6H_2O$), and chromium nitrate nonahydrate ($Cr(NO_3)_3 \cdot 9H_2O$) were added into the solution under constant stirring. Then, a proper amount of citric acid monohydrate ($C_6H_8O_7 \cdot H_2O$) was added as the complexing agent. After all the agents were entirely dissolved, ammonia solution (25% $NH_3$ by weight in water) was added to adjust the pH value of the solution to 7–9. After constant stirring at room temperature for a proper time, the mixed solution was heated on a heating platform until ignited. Then the product powder was collected and calcined at 700 °C in the air for 4 h to remove residual organic chemicals. The obtained catalyst precursor was in the gray-brown spongy powder form.

The LaCrO₃ support was synthesized by the same method as above. And the stoichiometric ratio of nickel nitrate hexahydrate (Ni(NO₃)₂•6H₂O), cobalt nitrate hexahydrate (Co(NO₃)₂•6H₂O) and LaCrO₃ support were added in deionized water (1.25 wt% Ni and 1.25 wt% Co of LaCrO₃ support). After stirring for 12 h, the mixture was dried at 80 °C for 12 h and then calcined at 700 °C in air for 4 h. The impregnated NiCo@LaCrO₃ was obtained.

**Characterization**. The crystalline phase structure of the catalyst samples was examined by an X-ray diffractometer (XRD, MXPAHF, MacScience) using Cu Kα radiation ($\lambda = 1.5406$ Å) over the range of $2\theta = 20-80°$ at room temperature.

X-ray photoelectron spectroscopy (XPS) analysis was performed using an electron spectrometer (ESCALAB 250, Thermo-VG Scientific, USA) with an exciting source of Al Kα = 1486.6 eV.

The microstructures of samples were observed by high-resolution transmission electron microscopy (HR-TEM, Talos F200X, FEI, USA) and high-angle annular dark-field scanning transmission electron microscopy (HAADF-STEM, JEM-ARM200F, JEOL, Japan) operating at an accelerating voltage of 200 kV. The element distribution was measured by energy-dispersive X-ray spectroscopy mapping analysis (EDS-Mapping, Talos F200X, FEI, USA).

Temperature programmed reduction (TPR) was carried out with a simultaneous thermal analyzer (STA449F3, NETZSCH, Germany). In all, 10–15 mg powder sample was placed in an alumina crucible and degassed at 230 °C in vacuum for 1 h to remove adsorbates. After cooled down to room temperature, the sample was heated in situ to 1200 °C with a heating rate of 10 °C min⁻¹. The flow rate of the forming gas (5 vol% H₂/N₂) was 60 sccm. We take the first-order derivative on the mass-loss curve as DTG.

Temperature programmed oxidation (TPO) was performed on the used catalysts to analyze the carbon deposition. The analysis was carried out with a simultaneous thermal analyzer (STA 449 F3, NETZSCH, Germany). 10–15 mg powder sample was placed in an alumina crucible. The sample was first heated to 800 °C under 10 sccm N₂ protection with a heating rate of 10 °C min⁻¹ to remove adsorbed gas molecules and to decompose any possible La₂O₂(CO₃). After cooled down to room temperature, the sample was heated to 1000 °C in dry air (flow rate = 10 sccm) with a heating rate of 10 °C min⁻¹. The mass loss detected in the high-temperature stage above 500 °C reflects the amount of deposited carbon.

The BET-specific surface areas were measured by nitrogen adsorption at liquid nitrogen temperature (77 K) using a surface area analyzer (NOVA 3200e, Quantachrome, USA). Before N₂ adsorption, the samples were degassed at 300 °C for 3 h to remove any residual moisture and other volatiles.

**Catalytic activity tests**. In all, 300 mg sample was placed in a fixed bed quartz reactor (i.d. = 6 mm) without dilution. The sample was activated in pure H₂ at 700 °C for 1 h before catalytic tests. After purging by N₂ for 30 min, the reactor was heated to the test temperature to carry out the catalyst activity test under a continuous feed of approximate equimolecular CO₂/CH₄ mixture with a flow rate of 60 mL min⁻¹ without dilution. The same gaseous hourly space velocity (GHSV) of $1.2 \times 10^4$ mL g$_{cat}$⁻¹ h⁻¹ was kept throughout the test. The steady-state tests were performed under atmospheric pressure at 750 °C. The reaction products were analyzed by online gas chromatography (GC9790, FULI), and the flow rate of the tail gas was measured by a soap film flowmeter. The temperature-dependent test was performed by reducing the temperature from 850 to 600 °C with a temperature step of 50 °C. At each step, the temperature was stabilized for 10 min and four data points were collected in the succeeding 120 min. The conversions of CH₄ and CO₂, H₂/CO ratio, and selectivities of H₂ and CO are defined as:

$$\text{Conv CH}_4 = \frac{[\text{CH}_4]_{in} - [\text{CH}_4]_{out}}{[\text{CH}_4]_{in}} \times 100\%$$

$$\text{Conv CO}_2 = \frac{[\text{CO}_2]_{in} - [\text{CO}_2]_{out}}{[\text{CO}_2]_{in}} \times 100\%$$

$$\text{H}_2/\text{CO ratio} = \frac{[\text{H}_2]_{out}}{[\text{CO}]_{out}} \times 100\%$$

$$\text{Selectivity H}_2 = \frac{2 \times [\text{H}_2]_{out}}{[\text{CH}_4]_{in} - [\text{CH}_4]_{out}} \times 100\%$$

$$\text{Selectivity CO} = \frac{[\text{CO}]_{out}}{[\text{CH}_4]_{in} - [\text{CH}_4]_{out} + [\text{CO}_2]_{in} - [\text{CO}_2]_{out}} \times 100\%$$

where $[\text{CH}_4]_{in}$ and $[\text{CO}_2]_{in}$ denote the molar flow rates of the introduced CH₄ and CO₂, $[\text{CH}_4]_{out}$, $[\text{CO}_2]_{out}$, $[\text{H}_2]_{out}$, and $[\text{CO}]_{out}$ denote the molar flow rates of CH₄, CO₂, H₂, and CO in the tail gas.

**DFT computational method**. Vienna Ab Initio Simulation Package (VASP) was adopted for our spin-polarized density functional theory calculations[55]. The ion-electron interaction and exchange-correlation were described using the projector-augmented plane-wave (PAW) approach and the Perdew–Burke–Ernzerhof (PBE)

functional[56,57]. The plane-wave basis set with a cutoff energy of 400 eV was selected for the calculations. DFT+U correction (U$_{eff}$ = 3.7 eV and 3.5 eV for Cr and Co, respectively) is considered to treat the 3$d$ orbital electrons[58,59].

A $2 \times 2$ supercell slab with eight atom layers is established for LaCrO₃ 010 and 001 surfaces. Cr atoms are replaced with Ni or Co atoms depending on the specific configuration (Supplementary Figs. 18 and 19) and the topmost O atoms (O$_\alpha$) were removed to ensure the open of 3d orbitals of the catalytically active transition metal elements. The bottom two layers were fixed while all other atoms were relaxed until the maximum force was less than 0.02 eV Å⁻¹. The vacuum space along the projection direction of the top surface was more than 15 Å to avoid the interactions between period images. The surface Brillouin zone was sampled by a $2 \times 2 \times 1$ k points mesh.

## Data availability

The authors declare that all data supporting the findings of this study are available within the article and its supplementary information files, and from the corresponding author on request.

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

## Acknowledgements

This work was supported by the National Natural Science Foundation of China (Grant no.: 21427804).

## Author contributions

T.Z. performed all the synthesis, most of the structural and compositional characterizations, and the DRM tests. X.T. performed the specific surface area and TPO characterization, and some DRM tests. H.Y. carried out some of the TPR tests and assisted in the data processing. M.L. assisted in the TPR and DRM tests. The paper was co-written by T.Z. and H.W. J. Zhao performed the DFT calculation and J. Zeng supervised DFT analysis. The research was supervised by H.W. All authors discussed the results and commented on the manuscript.

## Competing interests

The authors declare no competing interests.
