## [Peer Review File · Communications Chemistry]

Reviewers' comments:

Reviewer #1 (Remarks to the Author):

The manuscript presents the development of a highly active and thermostable submonolayer La(NiCo)O Δ catalyst stabilized by perovskite LaCrO₃ surface lattice for DRM reaction. This catalyst (LNCO) has higher activity and coking resistance than its monometallic (Ni or Co) counterparts for DRM reactions. The manuscript should be revised by considering the following issues before the publication. I vote for major revision.

1. In the initial 15h of the reaction (see Fig. 1(b)), LNCO has higher CH₄ and CO₂ conversion than LCO. This result may be related to the catalytic activity for DRM reactions. Therefore, it is necessary to present what properties of LNCO contributes to the high conversion and durability in long term reaction condition for better understanding about the catalytic activity using DFT calculations.
2. LNO and LCO showed a rapid decrease in the H₂/CO ratio during the reaction compared to LNCO. This result indicates that LNCO could suppress the RWGS reaction quite well. Authors can present more specific evidence about the reaction by presenting DFT calculation results such as energy profile about this reaction. This analysis may provide a better understanding of the higher H₂/CO ratio in LNCO suppressing the RWGS reaction.
3. As you mentioned, atomically dispersed NiO_x and CoO_x can contribute to enhancing the catalytic activity and coking resistance. The authors should clearly discuss this effect. For example, Ni-Co alloy has been reported to enhance activity and coking resistance by ligand and strain effects. Therefore, it is necessary to present the atomic scale analysis to explain what effect the interaction between Ni and Co enhanced the DRM reaction and coking resistance.
4. The authors said that the presence of monolayer NiO_x and CoO_x supported on the LaCrO₃ is responsible for the enhanced DRM activity and coking resistance. However, the reviewer doubt on the author's claim. The reduction process of catalyst and DRM condition can activate the segregation of cationic Ni or Co atom into surface layer, leading to the formation of metallic subnanometer Ni or Co clusters (so-called exsolution process). The authors should discuss the possible presence of metallic Ni or Co particles, which can significantly enhance CH₄ and CO₂ decomposition. Up to now, there has been no report that the NiO_x or CoO_x oxide system can improve DRM. The authors also experimentally showed the presence of large metallic Ni or Co particles, which was ignored by the authors as the possible reason to enhanced DRM activity.
5. The authors should discuss the possible formation of Ni-Co alloy.
6. The authors should present the possible presence of subnanometer Ni or Co or Ni-Co alloy particles, which is hard to detect experimentally.
7. The reviewer strongly think that the presence of metallic Ni or Co particles in LNCO catalyst is responsible for the enhanced DRM activity instead of the NiO_x and CoO_x monolayer. The authors should clearly discuss this point.

Reviewer #2 (Remarks to the Author):

The authors found a good mixed oxide catalyst for dry reforming of methane that is quite stable. They further characterized their catalyst with HR-TEM and proposed a model for its structure. Although the performance of their catalyst is impressive, I do find several issues with their paper.

1. The authors seemed to have mixed up messages for their catalyst: they motivated their work by single-atom catalysts, but claimed what they have is submonolayer/support type. I think claiming their catalyst as single atom is a far stretch.
2. TEM images are 2D projection. Unless they can clearly see the gas-solid interface, I don't think they can convincingly argue for the state of submonolayer on a support from the images they showed. They should test a control system by preparing the LaCrO₃ support first and then depositing Ni/CO ions by incipient wetness impregnation.
3. They claimed Ni-Co synergistic effect, but this needs to be demonstrated by experimental evidence such as EXAFS showing close distance.

We thank the reviewers for the constructive comments. We have addressed all the comments from the reviewers pointwise and revised our manuscript accordingly. The revised contents in the manuscript are marked by **red fonts** for clarity.

Reviewer #1 (Remarks to the Author):

The manuscript presents the development of a highly active and thermostable submonolayer $\text{La}(\text{NiCo})\text{O}_\Delta$ catalyst stabilized by perovskite LaCrO_3 surface lattice for DRM reaction. This catalyst (LNCO) has higher activity and coking resistance than its monometallic (Ni or Co) counterparts for DRM reactions. The manuscript should be revised by considering the following issues before the publication. I vote for major revision.

Comment 1:

1. In the initial 15h of the reaction (see Fig. 1(b)), LNCO has higher CH_4 and CO_2 conversion than LCO. This result may be related to the catalytic activity for DRM reactions. Therefore, it is necessary to present what properties of LNCO contributes to the **high conversion** and **durability** in long term reaction condition for better understanding about the catalytic activity using DFT calculations.

Answer 1:

(1) The stability of SML:

Thanks to the reviewer's constructive suggestion for the DFT calculation. We carried out DFT calculations for a few typical configurations of LNCO. The calculation results show that the energy of the system with neighbor Ni and Co atoms on the top layer (010 and 001 surface) is lower than other configurations. This result suggests that the interaction between neighboring Ni and Co atoms on the top layer can stabilize the submonolayer structure. Thus, the DFT

calculation supports our discussions about the stability of LNCO and explains the long-term durability of LNCO in the DRM reaction conditions. For details, please see the revised manuscript (page 25 paragraph 2, marked by red fonts) and Supplementary Information.

(2) The high conversions:

Yes, the LNCO catalyst has higher CH₄ and CO₂ conversions, and therefore, higher catalytic activity than LCO over the whole 100 h time on stream. In fact, not only LNCO but also LNO shows higher activity than LCO, especially, in the initial 15 h of the DRM reaction (see Fig. 1b). This is because both LNO and LNCO are Ni-containing while LCO is only Co-containing. As mentioned in the manuscript (page 4 paragraph 3), the catalytic activity of the submonolayer catalysts comes from the open 3d electron orbitals of the transition metals (Ni, or Co). It is well documented in the published literature that Ni has higher catalytic activity than Co for DRM reaction (Yentekakis, Panagiotopoulou et al. 2021). Thus, Ni atoms in LNCO should play a crucial role in the high conversions of CH₄ and CO₂. We have addressed this point in the revised manuscript (page 24 paragraph 1, marked by red fonts).

The key point of this work is that we found the novel perovskite-based submonolayer/support catalyst performs well for DRM: it can be highly active and thermostable. We explained the high activity of the oxide-based submonolayer catalyst based on well-established knowledge from the published papers. Comprehensive DFT calculation regarding the activity of LNCO deserves independent work and is beyond the scope of this manuscript.

Comment 2:

LNO and LCO showed a rapid decrease in the H₂/CO ratio during the reaction compared to LNCO. This result indicates that LNCO could suppress the RWGS

reaction quite well. Authors can present more specific evidence about the reaction by presenting DFT calculation results such as energy profile about this reaction. This analysis may provide a better understanding of the higher H₂/CO ratio in LNCO suppressing the RWGS reaction.

Answer 2:

Fig. 1a and Supplementary Fig. 1a show that the CH₄ and CO₂ conversions over LNCO are in thermo-equilibrium. Meanwhile, the H₂/CO ratio (0.9) of LNCO is also very close to the thermo-equilibrium value (0.92) (Pakhare and Spivey 2014). It is seen from Fig. 1b that the decrease in the H₂/CO ratio of LNO and LCO happened simultaneously with the decrease in the CH₄ and CO₂ conversions. The decrease of the CH₄ and CO₂ conversions over LNO and LCO to below the thermo-equilibrium value indicates the DRM reaction switches from the thermodynamic-control process into a kinetic-controlled process. It is reported that Ni is highly active for RWGS reaction, and the RWGS is very fast. It immediately reaches thermo-equilibrium in a wide temperature range (e.g. above 500 °C, depending on the contact time, etc.) (Wheeler 2004, Ponugoti and Janardhanan 2020). Or in other words, the RWGS is less sensitive to the catalysts' activity at high temperatures. In our experimental conditions (750 °C), the rapid decrease in the H₂/CO ratio of LNO and LCO should be dominated by the degradation of the catalyst, which reduces the amount of DRM-generated H₂ and CO. In this case, the less catalyst-dependent RWGS reaction will consume a relatively larger proportion of the total H₂ and add a more significant amount of CO to the total. Thus, LNO and LCO show a rapid decrease in the H₂/CO ratio during the reaction. On the other hand, the LNCO could suppress the RWGS reaction very well is because the catalyst is stable and the conversion of CH₄ and CO₂ to H₂ and CO maintains at a high level (see revised manuscript page 7 paragraph 1, marked by red fonts).

Comment 3:

As you mentioned, atomically dispersed NiOx and CoOx can contribute to enhancing the **catalytic activity** and **coking resistance**. The authors should clearly discuss this effect. For example, Ni-Co alloy has been reported to enhance activity and coking resistance by ligand and strain effects. Therefore, it is necessary to present the atomic scale analysis to explain what effect the interaction between Ni and Co enhanced the **DRM reaction** and **coking resistance**.

Answer 3:

As the reviewer noticed, we think the atomically dispersed Ni and Co (embedded in NiOx and CoOx) can contribute to enhancing the catalytic activity and coking resistance.

(1) Regarding the activity:

It has been proved that perovskites can have high catalytic activity for many reactions. For example, Suntvich et al. reported that perovskite $\text{Ba}_{0.5}\text{Sr}_{0.5}\text{Co}_{0.8}\text{Fe}_{0.2}\text{O}_{3-\delta}$ (BSCF) catalyzes the OER with intrinsic activity that is at least an order of magnitude higher than that of the state-of-the-art iridium oxide catalyst in alkaline (Suntivich, May et al. 2011). They proposed that the e_g orbital of surface transition metal ions participates in σ -bonding with a surface-anion adsorbate, and the e_g filling can greatly influence the binding of reaction intermediates to the oxide surface and thus the activity. Kim et al. (Kim, Qi et al. 2010) demonstrated that $\text{La}_{0.9}\text{Sr}_{0.1}\text{CoO}_3$ has higher activity than a commercial Pt-based catalyst for NO oxidation. It is also reported that perovskite $\text{La}_{0.75}\text{Sr}_{0.25}\text{Cr}_{0.5}\text{Mn}_{0.5}\text{O}_{3-\delta}$ (LSCM) can be used as an anode material for solid oxide fuel cells and shows high activity for CH_4 oxidation (Ge, Chan et al. 2012). In our case, we proposed that the high activity of LNCO comes from the atomically dispersed low-valent $\text{Ni}^{\delta+}$ (and $\text{Co}^{\delta+}$) metal cations with open 3d orbitals and filled e_g electrons. Considering that Ni is more active than Co for

DRM, we think the catalytic activity of LNCO is dominated by Ni, while Co mainly contributes to enhancing the stability of the submonolayer structure. Please see the revised manuscript (page 3, 4 and 24, marked by red fonts).

(2) Regarding the coking resistance:

Kim et al. (Kim, Suh et al. 2000) studied the effect of Ni particle size on coking for DRM reaction and found that Ni particles smaller than 7 nm can effectively suppress the formation of filamentous carbon and highly dispersed metal particles showed a remarkably low coking rate. Guo et al. reported that the absence of adjacent active metal atoms (Ni or Co) could prevent the C-C coupling or coking (Guo, Fang et al. 2014). It is also reported that Ni SACs favor partial CH₄ dehydrogenation and thus the complete decomposition to C is suppressed (Akri, Zhao et al. 2019). The higher oxygen affinity of Co favors enhancing the coking resistance (Wang, Dong et al. 2019). We discussed the coking resistance of LNCO by citing the above papers. In addition, we also proposed that the active oxygen (O_β) should be exchangeable with the oxygen species deduced from CO₂, and helps activate CO₂ and suppress carbon deposition. (see the Discussion Section in the revised manuscript)

Comment 4:

The authors said that the presence of monolayer NiO_x and CoO_x supported on the LaCrO₃ is responsible for the enhanced DRM activity and coking resistance. However, the reviewer doubt on the author's claim. The reduction process of catalyst and DRM can activate the segregation of cationic Ni or Co atom into surface layer, leading to the formation of metallic subnanometer Ni or Co clusters (so-called exsolution process). The authors should discuss the possible presence of metallic Ni or Co particles, which can significantly enhance CH₄ and CO₂ decomposition. **Up to now, there has been no report that the NiO_x or CoO_x oxide system can improve DRM.** The authors also experimentally showed the presence of large metallic Ni or Co particles, which

was ignored by the authors as the possible reason to enhanced DRM activity.

Answer 4:

- (1) As we mentioned in the manuscript, the high catalytic activity and very stable long-term performance differ the LNCO catalyst from the usual metal-support catalysts reported in literatures, especially if we consider the relatively low Ni loading (1.3 wt%) and low specific surface area (10.0 m²/g) of LNCO-F.
- (2) As revealed in Supplementary Fig. 9b, the Ni-Co metal nanoparticles in used LNCO are mainly very large ones (> 50 nm). The existence of large metal particles implies a weak metal-support interaction, which is not in favor of the stable existence of subnanometer Ni-Co particles.
- (3) The degradation behavior of LNO and LCO is different from that of metal-support catalysts. The conversions of CH₄ and CO₂ over LNO and LCO catalysts decrease in an accelerated way with time on stream. For comparison purposes, we have also fabricated a metal-support catalyst NiCo@LaCrO₃ of the same composition with LNCO by impregnation method. In contrast to the LNO and LCO, the decrease in conversions of CH₄ and CO₂ over NiCo@LaCrO₃ shows a slowdown trend (See Supplementary Fig. 2), which is in accord with the sintering degradation behavior of metal-support catalysts (Hansen, Delariva et al. 2013). But such a metal-support catalyst is less stable than LNCO.
- (4) Large metallic particles were observed in all the used LNCO, LNO, and LCO (see Supplementary Fig. 9, 12-14), but the final CH₄ and CO₂ conversions over deactivated LNO and LCO are very small. Therefore, we don't think the enhanced DRM activity of LNCO can be explained by the large and few metallic Ni-Co nanoparticles.
- (5) We have evidence of the 2D submonolayer structure, for example, see Fig. 4-6 and the discussions. The HRTEM and HAADF-STEM images show that the size of SML is several nanometers and the surface density of SML is

much higher than the big metallic particles. The LNCO-SML shares the same lattice with the perovskite LaCrO_3 support. If these SML are subnanometer Ni or Co, we should find the protrusion of Ni or Co in the gas-solid interface in the TEM images. But we cannot find any protrusions of 3D metal particles.

(6) DFT calculation results support the existence of NiO_x and CoO_x .

(7) We agree with the reviewer's comment that there has been no report that the NiO_x or CoO_x oxide system can improve DRM. We think this is mainly because NiO_x or CoO_x is in general not stable in Ni- or Co-containing perovskites. In the present work, we stabilized the NiO_x and/or CoO_x submonolayers by a stable irreducible perovskite LaCrO_3 .

We have addressed the above point in the revised manuscript (page 15)

Comment 5:

The authors should discuss the possible formation of Ni-Co alloy.

Answer 5:

We have observed the formation of Ni-Co alloy and discussed it in the manuscript. We provided the TEM and EDS-Mapping of NiCo alloy in Supplementary Fig. 9b. We can find that the surface density of NiCo alloy is low and the size of NiCo alloy is typically larger than 50 nm. Such few and large NiCo alloy cannot be responsible for the high catalytic activity of LNCO.

Please also refer to our responses to comment 4.

Comment 6:

The authors should present the possible presence of subnanometer Ni or Co or Ni-Co alloy particles, which is hard to detect experimentally.

Answer 6:

We searched but failed to find any evidence for the presence of subnanometer Ni or Co or Ni-Co alloy particles by using HRTEM and HAADF-STEM. Instead, we have evidence for the presence of Ni- and Co-containing submonolayers.

Comment 7:

The reviewer strongly think that the presence of metallic Ni or Co particles in LNCO catalyst is responsible for the enhanced DRM activity instead of the NiO_x and CoO_x monolayer. The authors should clearly discuss this point.

Answer 7:

- (1) The presence of large and few metallic Ni-Co particles in LNCO cannot explain the high activity of LNCO and the degradation of LNO and LCO. We cannot find any subnanometer Ni-Co nanoparticles with HRTEM and HAADF.
- (2) We have evidence for the presence of submonolayers.
- (3) Our DFT calculation results support the existence of submonolayers.
- (4) It has been proved that perovskites can have high catalytic activity for many reactions.
- (5) Please also refer to our responses to comment 3 and 4.

Reviewer #2 (Remarks to the Author):

The authors found a good mixed oxide catalyst for dry reforming of methane that is quite stable. They further characterized their catalyst with HR-TEM and proposed a model for its structure. Although the performance of their catalyst is impressive, I do find several issues with their paper.

Comment 1:

The authors seemed to have mixed up messages for their catalyst: they motivated their work by single-atom catalysts, but claimed what they have is submonolayer/support type. I think **claiming their catalyst as single atom** is a far stretch.

Answer 1:

The present work was inspired by single-atom catalysts. We believe the catalyst reported in this work is submonolayer/support type but we didn't intend to claim it as single-atom catalyst. The catalytic active Ni and Co atoms embedded in the oxide submonolayer are separated by O and La atoms and don't have any adjacent Ni or Co to form metallic bonds. So the active metal atoms in LNCO submonolayer are atomically dispersed. We have revised the manuscript to avoid any misleading of our catalyst as the single-atom catalyst. (see page 1 and 27, marked by red fonts).

Comment 2:

TEM images are 2D projection. Unless they can clearly see the gas-solid interface, I don't think they can convincingly argue for the state of submonolayer on a support from the images they showed. They should test a control system by preparing the LaCrO_3 support first and then depositing Ni/Co ions by incipient wetness impregnation.

Answer 2:

(1) We can clearly see the gas-solid interface in the HRTEM and HAADF images of LNCO, but we can't find any 3D feature. A 3D metal particle will appear as an observable protrusion in these images, but the 2D submonolayer is different. For example, in the following figure, we can easily find many protrusions of 3D nanoparticles in a metal-support type Ru/ LaCrO_3 sample. In contrast, we cannot see any protrusion in the HAADF

images of LNCO, even the lateral size of the 2D submonolayer region is similar to that of the Ru nanoparticles.

In fact, we proposed the submonolayer/support type catalyst in the present work is because the experimental results cannot be explained by the usual metal-support type catalysts. Please refer to our responses to comment 4 of Reviewer #1.

Figure 1 HAADF-STEM and HR-TEM figures with 2-5 nm Ru nanoparticles supported on LaCrO₃ (a) and LNCO-SML (b), for comparison purposes only.

(2) We synthesized NiCo@LaCrO₃ with the same nominal composition to

LNCO by preparing the LaCrO_3 support first and then depositing Ni/Co ions by incipient wetness impregnation. However, the catalytic performance is very different from LNCO, LNO and LCO prepared by the sol-gel combustion method. The conversions of CH_4 and CO_2 over NiCo@LaCrO_3 are lower than LNCO and decrease fast in a slowdown trend (see Supplementary Fig. 2), which is in accord with the sintering degradation behavior of metal-support catalysts (Hansen, Delariva et al. 2013). As a comparison, the conversions of CH_4 and CO_2 over LNO and LCO catalysts decrease in an accelerated way with time on stream. Such a NiCo@LaCrO_3 metal-support catalyst is less stable than LNCO.

The TEM and EDS-Mapping images of NiCo@LaCrO_3 after 10 h time on stream were also added in the revised Supplementary Information (see Supplementary Fig. 15). No submonolayer can be observed in Supplementary Fig. 15.

We have added the result of NiCo@LaCrO_3 and discussed it in the revised manuscript. (Please see page 8).

Comment 3:

They claimed Ni-Co synergistic effect, but this needs to be demonstrated by experimental evidence such as EXAFS showing close distance.

Answer 3:

We tried to carry out the EXAFS characterization (Shanghai Synchrotron Radiation Facility) for reduced $\text{LaNi}_{0.05}\text{Co}_{0.05}\text{Cr}_{0.9}\text{O}_3$, reduced impregnated NiCo@LaCrO_3 , and fresh $\text{LaNi}_{0.05}\text{Co}_{0.05}\text{Cr}_{0.9}\text{O}_3$. But we failed to get any meaningful EXAFS data because the Ni or Co content is below the detection limit.

Instead of seeking evidence from EXAFS, we carried out DFT calculations for a few typical configurations of LNCO. The calculation results show that the energy of the system with neighbor Ni and Co atoms on the top layer (010 and 001 surface) is lower than other configurations. This result suggests that the interaction between neighboring Ni and Co atoms on the top layer can stabilize the submonolayer structure. Thus, the DFT calculation supports our discussions about the stability of LNCO and explains the long-term durability of LNCO in the DRM reaction conditions. Please see the revised manuscript (page 25 paragraph 2, marked by red fonts) and Supplementary Information.

References

- Akri, M., S. Zhao, X. Li, K. Zang, A. F. Lee, M. A. Isaacs, W. Xi, Y. Gangarajula, J. Luo, Y. Ren, Y.-T. Cui, L. Li, Y. Su, X. Pan, W. Wen, Y. Pan, K. Wilson, L. Li, B. Qiao, H. Ishii, Y.-F. Liao, A. Wang, X. Wang and T. Zhang (2019). "Atomically dispersed nickel as coke-resistant active sites for methane dry reforming." Nature Communications **10**(1): 5181.
- Ge, X.-M., S.-H. Chan, Q.-L. Liu and Q. Sun (2012). "Solid Oxide Fuel Cell Anode Materials for Direct Hydrocarbon Utilization." Advanced Energy Materials **2**(10): 1156-1181.
- Guo, X., G. Fang, G. Li, H. Ma, H. Fan, L. Yu, C. Ma, X. Wu, D. Deng, M. Wei, D. Tan, R. Si, S. Zhang, J. Li, L. Sun, Z. Tang, X. Pan and X. Bao (2014). "Direct, nonoxidative conversion of methane to ethylene, aromatics, and hydrogen." Science **344**(6184): 616-619.
- Hansen, T. W., A. T. Delariva, S. R. Challa and A. K. Datye (2013). "Sintering of catalytic nanoparticles: particle migration or Ostwald ripening?" Acc Chem Res **46**(8): 1720-1730.
- Kim, C. H., G. Qi, K. Dahlberg and W. Li (2010). "Strontium-doped perovskites rival platinum catalysts for treating NO_x in simulated diesel exhaust." Science **327**(5973): 1624-1627.
- Kim, J.-H., D. J. Suh, T.-J. Park and K.-L. Kim (2000). "Effect of metal particle size on coking during CO₂ reforming of CH₄ over Ni-alumina aerogel catalysts." Applied Catalysis A: General **197**(2): 191-200.
- Pakhare, D. and J. Spivey (2014). "A review of dry (CO₂) reforming of methane over noble metal catalysts." Chemical Society Reviews **43**(22): 7813-7837.

- Ponugoti, P. V. and V. M. Janardhanan (2020). "Mechanistic Kinetic Model for Biogas Dry Reforming." Industrial & Engineering Chemistry Research **59**(33): 14737-14746.
- Suntivich, J., K. J. May, H. A. Gasteiger, J. B. Goodenough and Y. Shao-Horn (2011). "A perovskite oxide optimized for oxygen evolution catalysis from molecular orbital principles." Science **334**(6061): 1383-1385.
- Wang, H., X. Dong, T. Zhao, H. Yu and M. Li (2019). "Dry reforming of methane over bimetallic Ni-Co catalyst prepared from $\text{La}(\text{Co}_x\text{Ni}_{1-x})_{0.5}\text{Fe}_{0.5}\text{O}_3$ perovskite precursor: Catalytic activity and coking resistance." Applied Catalysis B: Environmental **245**: 302-313.
- Wheeler, C. (2004). "The water-gas-shift reaction at short contact times." Journal of Catalysis **223**(1): 191-199.
- Yentekakis, I. V., P. Panagiotopoulou and G. Artemakis (2021). "A review of recent efforts to promote dry reforming of methane (DRM) to syngas production via bimetallic catalyst formulations." Applied Catalysis B: Environmental **296**: 120210.

REVIEWERS' COMMENTS:

Reviewer #1 (Remarks to the Author):

The authors have revised the manuscript as suggested by the reviewers. Thus, I agree with the publication in current form.